

# Whole genome sequencing of group A *Streptococcus*: development and evaluation of an automated pipeline for *emm* gene typing

Georgia Kapatai[1], Juliana Coelho[1], Steven Platt[2] and Victoria J. Chalker[1]

[1] Respiratory and Vaccine Preventable Bacterial Reference Unit, Public Health England, London, United Kingdom
[2] Infectious Disease Informatics, Public Health England, London, United Kingdom

Corresponding author
Georgia Kapatai,
georgia.kapatai@phe.gov.uk

## ABSTRACT

*Streptococcus pyogenes* group A *Streptococcus* (GAS) is the most common cause of bacterial throat infections, and can cause mild to severe skin and soft tissue infections, including impetigo, erysipelas, necrotizing fasciitis, as well as systemic and fatal infections including septicaemia and meningitis. Estimated annual incidence for invasive group A streptococcal infection (iGAS) in industrialised countries is approximately three per 100,000 per year. Typing is currently used in England and Wales to monitor bacterial strains of *S. pyogenes* causing invasive infections and those isolated from patients and healthcare/care workers in cluster and outbreak situations. Sequence analysis of the *emm* gene is the currently accepted gold standard methodology for GAS typing. A comprehensive database of *emm* types observed from superficial and invasive GAS strains from England and Wales informs outbreak control teams during investigations. Each year the Bacterial Reference Department, Public Health England (PHE) receives approximately 3,000 GAS isolates from England and Wales. In April 2014 the Bacterial Reference Department, PHE began genomic sequencing of referred *S. pyogenes* isolates and those pertaining to selected elderly/nursing care or maternity clusters from 2010 to inform future reference services and outbreak analysis ($n = 3,047$). In line with the modernizing strategy of PHE, we developed a novel bioinformatics pipeline that can predict *emm* types using whole genome sequence (WGS) data. The efficiency of this method was measured by comparing the *emm* type assigned by this method against the result from the current gold standard methodology; concordance to *emm* subtype level was observed in 93.8% (2,852/3,040) of our cases, whereas in 2.4% ($n = 72$) of our cases concordance was observed to *emm* type level. The remaining 3.8% ($n = 117$) of our cases corresponded to novel types/subtypes, contamination, laboratory sample transcription errors or problems arising from high sequence similarity of the allele sequence or low mapping coverage. De novo assembly analysis was performed in the two latter groups ($n = 72 + 117$) and was able to diagnose the problem and where possible resolve the discordance (60/72 and 20/117, respectively). Overall, we have demonstrated that our WGS *emm*-typing pipeline is a reliable and robust system that can be implemented to determine *emm* type for the routine service.

## INTRODUCTION

Group A *Streptococcus* (GAS) or *Streptococcus pyogenes* is a human pathogen causing infections ranging from mild bacterial throat infection to severe septicaemia and meningitis (*Cunningham, 2000*). Invasive GAS infections (iGAS), though relatively uncommon compared to highly prevalent non-invasive GAS infections, are a significant global cause of morbidity and mortality. An increase in the incidence rates of iGAS in the last two decades (*Cunningham, 2000*; *Meehan et al., 2013*; *Guy et al., 2014*) has led to the introduction of national enhanced surveillance protocols in a number of developed countries, including the UK (*Lamagni & Williams, 2009*). In England and Wales, multiple outbreaks of *S. pyogenes* infection occur each year in locations such as schools, care homes, hospitals and family clusters. Sequence analysis of the *emm* gene is the main method used to aid bacterial discrimination and inform epidemiological study of group A streptococcal clusters and monitor the prevalence of types nationally within the population.

The *emm* gene encodes for the M-protein, a surface protein and a major virulence factor in GAS (*Sanderson-Smith et al., 2014*). The N-terminus hypervariable region of the M-protein is the source of its antigenic diversity and the targeted region for *emm* gene sequence typing (*Beall, Facklam & Thompson, 1996*; *Facklam et al., 1999*). Currently there are more than 200 *emm* types described (*McMillan et al., 2013*), but only a small proportion of these have been validated for the expression of the M-antigen (*Denny & Perry, 1957*; *Lancefield, 1959*).

The recent advances in whole genome sequencing technologies resulted in reduced costs and reduced turnaround times making this technology accessible to reference microbiology. Whole genome sequencing (WGS) is not just an alternative to Sanger sequencing but can offer increased resolution and higher predictive value for *emm* typing as demonstrated by *Athey et al. (2014)*. Here we describe the implementation and validation of a novel WGS-based *emm* typing tool within a reference microbiology lab for a large dataset of GAS isolates ($n = 3,047$).

## MATERIALS AND METHODS

### Isolates

Reference strains ($n = 10$) for all serotypes were acquired from PHE archives whereas 3047 clinical isolates were collected and sequenced prospectively over a period of 14 months, between April 2014 and May 2015. The FASTQ files for the isolates described in *Athey et al. (2014)* ($n = 191$) were obtained from the European Nucleotide Archive (ENA) study PRJNA233611.

### Microbiology

*Streptococcus pyogenes* isolates were cultured using standard methods (*Johnson et al., 1996*). The Public Health England National Streptococcal Reference Laboratory (Bacteriology

Reference Department) performed *emm* gene sequence typing on referred isolates obtained as previously described (*Podbielski, Melzer & Lütticken, 1991*; *Beall, Facklam & Thompson, 1996*) using a crude DNA extract for PCR and Sanger sequencing. In brief, the *emm* types were determined according to the protocol and guidelines available on the CDC website (https://www.cdc.gov/streplab/protocol-emm-type.html). When sequence data obtained using the CDC recommended primers generate ambiguous sequence, alternative primers (MF1, 59-ATAAGGAGCATAAAAATGGCT-39, and MR1, 59-AGCTTAGTTTTCTTCTTTGCG-39) (*Podbielski, Melzer & Lütticken, 1991*) (Sigma-Aldrich, St. Louis, MO, USA) were used for the amplification of the *emm* gene (*Podbielski, Melzer & Lütticken, 1991*). For whole genome sequencing preparation, purified DNA was prepared by using the QIAsymphony SP automated instrument (Qiagen, Hilden, Germany) and QIAsymphony DSP DNA Mini Kit, using the manufacturer's recommended tissue extraction protocol for Gram positive bacteria (including a 1 h pre-incubation with mutanolysin and lysozyme followed by 2 h incubation with proteinase K in ATL buffer and RNAse A treatment). DNA concentrations were measured using the Quant-iT dsDNA Broad-Range Assay Kit (Life Technologies, Paisley, UK) and GloMaxR 96 Microplate Luminometer (Promega, Southampton, UK). A Nextera XT DNA Library Preparation Kit (Illumina, San Diego, CA, USA) was used followed by sequencing using a HiSeq 2500 System (Illumina) and the $2 \times 100$-bp paired-end mode.

## Bioinformatic processing

Casava 1.8.2 (Illumina inc. San Diego, CA, USA) was used to deplex the samples and FASTQ reads were processed with Trimmomatic (*Bolger, Lohse & Usadel, 2014*) to remove bases from the trailing end that fall below a PHRED score of 30. Processed FASTQ reads from all sequences in this study were submitted to ENA using the ena_submission tool (https://github.com/phe-bioinformatics/ena_submission) and can be found at the PHE Pathogens BioProject PRJEB17673 at ENA (http://www.ebi.ac.uk/ena/data/view/PRJEB17673; Table S1).

K-mer identification software (https://github.com/phe-bioinformatics/kmerid) was used to compare the sequence reads with a panel of curated NCBI RefSeq genomes to identify the species. A sample of k-mers (DNA sequences of length k) in the sequence data are compared against the k-mers of 1,769 reference genomes representing 59 pathogenic genera obtained from RefSeq (NCBI Reference Sequence Database). The reference genome containing the most k-mers found in the sample is identified, and provides initial confirmation of the species. This step also identifies samples containing more than one species of bacteria (i.e., mixed cultures) and any bacteria misidentified as *S. pyogenes* by the sending laboratory. Further analysis continued only if *S. pyogenes* was identified.

## *Emm* gene typing tool implementation

The *emm* gene typing tool assigns *emm* type and subtype by querying the CDC M-type specific database (ftp://ftp.cdc.gov/pub/infectious_diseases/biotech/tsemm/). An updated version of the database is downloaded on a weekly base and genomic reads are mapped

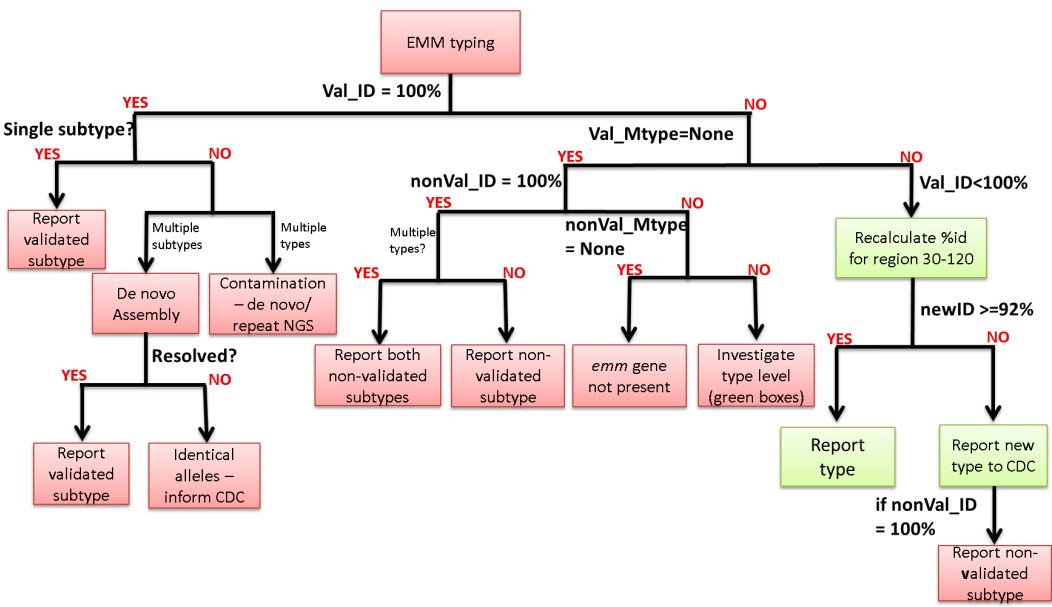

**Figure 1  Decision algorithm for the assignment of *emm* type/subtype.** The decision algorithm is based on the CDC guidelines that clinical scientists currently follow for assigning *emm* type available at http://www.cdc.gov/streplab/assigning.html.

to the latest version using bowtie2 (version 2.1.0; following options used: –fr –no-unal –minins 300 –maxins 1100 -k 99999 -D 20 -R 3 -N 0 -L 20 -I S,1,0.50) (*Langmead & Salzberg, 2012*). At this stage the validated *emm* genes (*emm*1–124; validated refers to confirmation of M protein presence using antiphagocytic tests (*Horstmann et al., 1988*; *Beall, Facklam & Thompson, 1996*; *Facklam et al., 1999*; *Facklam et al., 2002*)) were separated from the non-validated (*emm*125+, STC and STG) and both sets were analysed in parallel. In each set, alleles with 100% coverage (minimum depth of 5 reads per bp) over the length of their sequences and >90% identity were selected and the allele with the highest percentage identity was reported. Following this selection, a decision algorithm (Fig. 1) is implemented to determine (a) whether the validated or not validated *emm* type will be reported, (b) whether *emm* type or subtype will be reported and (c) whether further investigation is necessary (i.e., contamination, new type or presence of two *emm* subtypes). Due to the stringent filters implemented by this tool, in some cases, low coverage at the 5′ end of the allele, resulted to the non-validated *emm* type being reported when in fact the validated *emm* type was present. To avoid this, such cases are marked with '**' and reported as 'Not determine' so that further investigation is instigated. The *emm*-typing tool is publicly available in GitHub (https://github.com/phe-bioinformatics/emm-typing-tool).

## Decision algorithm

The decision algorithm for the assignment of *emm* type/subtype follows the CDC guidelines (http://www.cdc.gov/streplab/assigning.html). An *emm* subtype is assigned only in cases of 100% identity over the entire 180 bp length of the allele sequence. The allele sequence corresponds to the first 150 bp of the *emm* gene plus 30 bp from the upstream sequence

corresponding to the signal peptide portion. If subtype cannot be assigned, the *emm* type can be assigned if >92% identity is observed over the length of bases 30–120 which correspond to the first 90 bp of the *emm* gene.

The first part of the decision algorithm works within the *emm* typing tool; it attempts to assign *emm* subtype/type using the validated result first and if a validated *emm* type cannot be assigned or no validated result is available then *emm* subtype or type is assigned based on the non-validated result. In cases where an *emm* type or subtype cannot be assigned, the decision workflow (Fig. 1) can help to determine the appropriate course of action based on the reported results for the validated and non-validated set; for example, if two *emm* types are reported in the validated set this indicates contamination and the sample should be re-extracted and re-sequenced, whereas if two *emm* subtypes are reported then the genomic data should be assembled and further analysis performed to resolve and assign the correct *emm* subtype.

### De novo assembly

Genomic reads were assembled using SPAdes (version 2.5.1) de novo assembly software (*Bankevich et al., 2012*) with the following parameters 'spades.py –careful -1 strain.1.fastq.gz -2 strain.2.fastq -t 4 -k 33,55,77,85,93'. The resulting contigs.fasta file was converted into a BLAST database using blast+ (version 2.2.27) (*Camacho et al., 2009*) and queried using selected query sequence (i.e., allele sequence).

## RESULTS

### Comparing Sanger sequencing and WGS-based *emm* typing results

Group A *Streptococcus* isolates ($n = 3,047$) collected over a period of 14 months, between April 2014 and May 2015, were sequenced and *emm* gene typing data derived from genomic analysis were compared to data derived using the traditional Sanger sequencing method. Six isolates were removed from the analysis due to low yield following whole genome sequencing. Of the remaining 3,041 isolates, 2852 (93.8%; 95% CI [92.9–94.6]%) were concordant to *emm* subtype level (matched over the full length of the allele; 180 bps), whereas 72 (2.4%; CI [1.8–2.9]%) were concordant to *emm* type level (matched over the region 30-120) and only 117 (3.8%; CI [3.2–4.6]%) were discordant (Table 1).

Of the 72 isolates concordant to *emm* type level, seven were assigned different subtypes using the two methods, one was assigned two non-validated *emm* types (*emm*159.0/*emm*246.0), one of which (*emm* 246.0) corresponded to the PCR-derived result and three were assigned subtype by WGS but were only assigned type by Sanger sequencing. Another isolate was originally typed as a mixed culture (*emm*3.1/*emm*12.0) using the traditional method and following WGS analysis on single colonies only type *emm*3.1 was detected. The remaining 60 were assigned subtype by Sanger sequencing but were only assigned type by whole-genome sequencing. Upon further investigation, it became apparent that the *emm* gene typing tool was unable to call subtype due to the presence of mixed subtypes ($n = 6$) or mixed bases ($n = 54$); these are positions where a non-reference nucleotide is present in 20–80% of the reads making it impossible to call a consensus nucleotide. Interestingly, 31/54 isolates with mixed bases were *emm*44 and

**Table 1  Comparison of *emm* typing results by Sanger and WGS-based sequencing.** Results shown for the 20 most common *emm* types (corresponding to >85% of the study dataset). Concordant to *emm* sub-type, *emm* type level, and discordant are represented as "TRUE", 'TRUE?" and "FALSE", respectively.

| M-type (Wet lab) | NGS analysis | | | | Grand Total |
|---|---|---|---|---|---|
| | Sample failure | TRUE | TRUE? | FALSE | |
| 1.0 | | 557 | | 17 | 574 |
| 3.1 | 1 | 413 | 6 | 21 | 441 |
| 12.0 | | 302 | 9 | 7 | 318 |
| 89.0 | | 296 | 2 | 9 | 307 |
| 28.0 | | 206 | 1 | 10 | 217 |
| 75.0 | 1 | 124 | 1 | 5 | 131 |
| 4.0 | 2 | 112 | | 2 | 116 |
| 3.93 | | 79 | 2 | 4 | 85 |
| 6.0 | | 77 | 1 | 2 | 80 |
| 87.0 | | 58 | | 3 | 61 |
| 94.0 | | 49 | | 1 | 50 |
| 11.0 | | 44 | 1 | | 45 |
| 2.0 | | 38 | 1 | 1 | 40 |
| 44.0 | | | 31 | 2 | 33 |
| 18.0 | 1 | 27 | | 1 | 29 |
| 5.23 | | 28 | | 1 | 29 |
| 81.0 | | 24 | | 2 | 26 |
| 6.4 | | 20 | | 1 | 21 |
| 73.0 | | 19 | | | 19 |
| 22.0 | | 18 | | | 18 |
| Other | 1 | 361 | 17 | 28 | **407** |
| **Grand Total** | **6** | **2,852** | **72** | **117** | **3,047** |

this constituted the total number of emm44 isolates available in this study, suggesting that there might be a correlation between this *emm* type and the presence of mixed subtype variants within patients; an observation that warrants further investigation. The 60 isolates with mixed signature and the six subtype discordances were analysed further using de novo assembly and BLAST analysis. This approach was able assign a single subtype for all 60 isolates with mixed signature, with the *emm* allele identified in the assembly matching the subtype previously detected by Sanger sequencing. In the case of the seven isolates, where different subtypes were called, this approach confirmed the original WGS result. Further investigation into the sequence trace files, revealed that in five of these isolates analysis was done on a shorter PCR amplicon, suggesting amplification bias.

Of the 117 discordant isolates, 89 were due to different *emm* types being called with the two methods, four were flagged as 'Not determined' by WGS, four were flagged as mixed *emm* types, four were non-typeable by Sanger but typed to *emm* subtype level with WGS and 16 were flagged as possible new type by WGS. All discordant isolates were analysed further using de novo assembly and BLAST analysis.

The 'mixed' flag is assigned when two validated *emm* types are present with 100% coverage and identity ($n = 4$) whereas 'Not determined' flag is assigned when two or more alleles are present with the same percent identity but <100% ($n = 1$). 'Not determine' is also assigned when the validated *emm* type has low coverage issues in the last 1–3 bases of the 5′ end. In this case *emm* type will be tagged with '**' ($n = 3$). De novo assembly analysis was able to confirm presence of multiple *emm* genes in the five mixed isolates (attributed to contamination) and assign the validated *emm* type in the remaining three isolates (Table S2).

Further investigation of the 16 isolates flagged as carrying *emm* gene sequences not currently in the database (new types; all *emm*3 type) by WGS revealed that in all cases low mapping coverage (failed coverage = 100% filter due to base coverage <5 reads in one or more bases) towards the end of the *emm* gene sequence detected by Sanger sequencing was responsible for this result. De novo assembly analysis confirmed the presence of the *emm* allele previously detected by Sanger sequencing for these 16 isolates (12/16 *emm*3.1).

In the four cases where isolates were non-typeable by Sanger sequencing, this was attributed to problems with PCR amplification and/or mixed traces from Sanger sequencing data; these were assigned an *emm* type by WGS suggesting that this method has increased ability to differentiate between *emm* and *emm*-like genes.

De novo analysis suggested that in 76 of the 89 isolates with different *emm* types being called with the two methods, discordance was due to laboratory sample transcriptional error (Table S3). In the remaining 13 cases the presence of two *emm* genes was confirmed; in eight cases, pairs of validated *emm* genes were found suggesting laboratory contamination, whereas in one case, a validated (*emm*75) and non-validated gene (*emm*170) were found with Sanger method calling the non-validated type and WGS correctly calling the validated type. In another case, WGS returned the non-validated gene due to low coverage of the validated *emm* gene (only one read covering the last 20 bps) despite the mechanism to tag low coverage issues; in this scenario coverage and depth metrics fall below the threshold for accepting an allele even though the allele is assembled by the de novo assembly method.

In another three cases, where the Sanger method was calling the non-validated and WGS was calling the validated type, de novo analysis revealed that the two genes (*emm*60/*emm*169.3, *emm*34/*emm* 230) were actually quite similar and when analysed using BLAST methodology against the assembly were found to map to the same region; in fact when aligning against the *emm* type defining region (30–120 bps) more than 92% identity was observed for the validated *emm* types. In this situation, WGS has correctly assigned the validated *emm* gene.

Finally, in four cases, isolates assigned as non typeable by Sanger sequencing due to problems of PCR amplification and/or mixed traces during Sanger sequencing, were assigned an *emm* type by WGS suggesting that this method has increased ability to differentiate between *emm* and *emm*-like genes (Table S3).

## De novo investigation of known and novel *emm/emm*-like pairs

As previously described by *Athey et al. (2014)*, the presence of *emm*-like genes in the CDC *emm* typing database, can in some cases complicate WGS-based *emm* typing. Our *emm* gene typing tool is able to resolve this by reporting the presence of validated and non-validated

**Figure 2  Schematic showing the chromosomal arrangement of *emm*/*emm*-like gene pairs.** Positional analysis using de novo assembly and BLAST analysis was performed on representatives of isolates with 100% coverage/identity of both *emm* and *emm*-like genes.

*emm* genes and using this approach it was possible to identify pairs of *emm*/*emm*-like genes by selecting those where both validated and non-validated *emm*-genes are present with 100% identity. Positional analysis in a subset of our isolates, using de novo assembly and BLAST confirmed the presence of both genes (Table S4). This analysis confirmed a number of previously described *emm*-like genes (*emm*170, *emm*159 and more) (*Athey et al., 2014*) and also identified novel ones (*emm*134 and *emm*167).

Further investigation into the position of the *emm*-like genes in relation to the *emm* gene revealed that *emm*-like genes can be found either before or after *emm* in gene regions previously described for *mrp* and *enn*, respectively (*Athey et al., 2014*) (Fig. 2). Most *emm*-

like genes were found either exclusively before or after *emm* suggesting that they could belong to the *mrp* or *enn* gene family. However, two *emm*-like genes (*emm*156.0 and *emm*205.0) have been found in both positions depending on the accompanying *emm* gene.

### Validation of the *emm* typing tool using Athey et al. isolates

The *emm* typing tool was further validated using the Canadian isolates used in *Athey et al. (2014)*. Following analysis with the *emm* typing tool the results were compared to the results previously reported. In 186/191 cases the results obtained in this study are concordant with the results obtained in Athey et al. (Table S5). However, since in Athey et al. the results were reported to *emm* type not subtype, the comparison is limited to *emm* type. De novo assembly was used to investigate the five discordant cases, In two cases our pipeline predicted *emm*60 whereas the earlier study predicted *emm*169.3; blast analysis revealed presence of both genes with 100% identity for *emm*169.3 but >92% identity for the *emm*60 defining region (30–120 bp), therefore based on the CDC guidelines *emm*60 type should be assigned. In a third case, *emm*138.0 was assigned in this study and *emm* 192 was assigned previously; blast analysis revealed presence of both genes with 100% identity for *emm*138.0 and 99% identity for *emm*192. In the fourth case, the sample previously assigned *emm*4 was now assigned *emm*236.3 and blast analysis revealed presence of both genes with 100% identity. Further investigation into the mapping approach revealed low depth (<5 reads) for position 180 of *emm*4. Finally, the fifth case was previously assigned *emm*14 but our approach detected a mixed sample (*emm*14.3/*emm*51.0); blast analysis confirmed presence of both genes with 100% identity (Table S5).

## DISCUSSION

The recent reduction in price and turnaround time for WGS, and the rapid development of bioinformatics infrastructures to analyse and store the large amount of data generated are making this technology accessible to reference microbiology (*Loman et al., 2012*). However, before this technology can be implemented, steps need to be taken to ensure backward compatibility with the current 'gold standard' typing methodologies and schemes. Unlike serology-based typing methods, where a change to WGS would entail a complete change in methodology (*Kapatai et al., 2016*), sequencing-based methods like MLST and *emm* typing only require a change of sequencing platform. Already, *Athey et al. (2014)* have shown that WGS can be used to derive *emm* type from genomic data with the right bioinformatics tools.

In this study we present a novel bioinformatics tool for WGS-based *emm* gene typing, that uses a mapping approach, incorporating the CDC *emm* typing database (source file updated weekly) as a reference, and a decision algorithm resembling the decision process currently used by clinical scientists in the lab. Our *emm* typing tool, uses the logic within the decision algorithm (Fig. 1) to differentiate between validated (*emm* genes) and non-validated (*emm* or *emm*-like genes) *emm* types and assign *emm* type with a precision that allows *emm* subtype reporting.

A cohort of 3047 GAS isolates, previously *emm* typed by Sanger sequencing, were submitted to genomic sequencing and analysed using the *emm* typing tool. Results

were collected from 3,041 isolates (six failed genomic sequencing) and following initial comparisons, concordance was observed for 2,852 isolates (93.8%) to *emm* subtype level, whereas for 72 (2.4%) isolates concordance was observed to *emm* type level. Following further investigation this *emm*-type/subtype discrepancy was attributed to the sensitivity of the mapping approach: for 60/72 cases the *emm* subtype was not called due to the presence of mixed bases in certain positions ($n = 54$) or mapping to more than one subtypes (tag: 'mixed subtypes'; $n = 6$) that prevented the assignment of a consensus base, and de novo assembly and BLAST analysis were able to correctly assign the *emm* subtype in relation to Sanger sequencing results. The presence of mixed bases could be due to the presence of multiple subtypes and may be lineage specific as the majority of isolates 31/54 in this study corresponded to *emm*44. In 7/72 cases, *emm* subtype discrepancies were due to different subtypes called with the two methods and de novo analysis was unable to resolve this as it confirmed the mapping-based WGS result. However, further investigation into the Sanger sequence files revealed an amplicon problem stemming from PCR amplification bias in 5/7 of these cases, indicating the original Sanger result is most likely to have been incorrect. One of the remaining 5/72 isolates was a confirmed mixed sample (*emm*3.1/*emm* 12.0) by Sanger sequencing where multiple ($n = 3$) single colonies were used for WGS. Unfortunately in this case all colonies were *emm*3.1 but this is due to the colonies picked and not a failure of the method; for a second mixed isolate (*emm*3.1/*emm*12.0) analysed using the same approach, WGS analysis reported two colonies *emm*3.1 and 1 colony *emm*12.0.

The remaining 117 discrepant isolates were discordant at different levels: 76 isolates had different *emm* types due to possible errors in laboratory labelling and de novo assembly confirmed the mapping result. In 13 cases de novo assembly identified presence of a mixed *emm* profile that is the presence of two validated *emm* types in the same genomic sequence. Mapping identified the *emm* type with the highest coverage and missed the second type due to incomplete locus coverage; four cases were non-typeable by Sanger sequencing due to the presence and preferential sequencing of non-specific PCR amplicons but *emm* subtype was reported by WGS; 16 cases were flagged as 'New type' by WGS but further analysis identified this as a result of low mapping coverage; four cases were flagged as 'Mixed sample' where two *emm* types are reported with 100% identity whereas four other cases were flagged as 'Not determined', as result of two or more validated *emm* alleles reported with the same percent identity (100% > id > 90% plus 100% coverage) ($n = 1$) or low mapping coverage at the 5′ end ($n = 3$). De novo analysis was able to confirm presence of multiple *emm* genes in the five mixed isolates and assigned the tagged ('**') *emm* type to the remaining three.

Furthermore, in order to compare this method with the previously described method (*Athey et al., 2014*), the isolates from Athey et al. where analysed using our *emm*-typing tool and in 186/191 cases where concordant (to *emm* type level since no subtype was provided in previous reference). The five discordant cases were investigated further using de novo assembly and two cases where due to the previously described *emm*60/*emm*169.3 issue, whereas one case Athey et al. reported the non-validated typed (*emm*192) whereas our tool detected presence of both alleles (*emm*138.0/*emm*192.0) but reported the validated type (*emm*138.0). In the remaining two cases, one was tagged as having low coverage issues and

de novo assembly assigned the previously reported type whereas the other was reported by mapping as mixed which was then confirmed by de novo. The two methods, although both using a mapping approach, differ in the scoring method for assigning top hit; whereas our approach uses coverage and identity to identify validate non-validated types, the approach described by Athey et al. uses the SRST2 scoring system that uses binomial testing (*Inouye et al., 2014*).

Overall, our analysis demonstrates that WGS can be used for *emm* typing; in 93.8% of the cases (2852 concordant + 1 single colony match to one of mixed *emm* types) the *emm* typing tool was able to assign the correct *emm* subtype unaided whereas in 2.6% of the cases (60 called to type, 3 'Not determined' and 16 'New type') de novo assembly was used to help assign the correct *emm* subtype. Using both methods, 96.4% concordance to the traditional Sanger sequencing results was observed. These cases highlighted some problems stemming from the sensitivity of mapping; mixed bases when the presence of an alternative bases in 20–80% of the reads hinders a consensus call for those positions and mixed subtypes when the two allele sequences share high similarity. The latter usually involves alleles where the downstream flanking sequence of the first allele is identical to part of the sequence of the second allele; therefore reads covering both of the alleles are present. In this case using de novo assembly can clearly shows the overlap between the two sequences and usually a gap in the one of the alleles. As demonstrated here, these issues can be resolved if de novo assembly is used and no repeat is necessary. In contrast problems with Sanger sequencing (PCR amplicon or sequence trace problems) usually require a repeat of the entire process ($n = 15$). In one case, allele *emm*170.0 was amplified and reported by Sanger sequencing when both mapping and de novo assembly demonstrated presence of both *emm*75.0 and *emm*170.0. These two alleles are frequently seen together in isolates from the UK and *emm*170.0 is a known *emm*-like gene (*Athey et al., 2014*); issues with these two alleles are common in our laboratory and usually seen as mixed chromatographs, which then necessitates the use of alternative primers to resolve. Our WGS approach has been specifically designed to report presence of validated and non-validated *emm* types and thus can detect presence of such *emm/emm*-like pairs.

In this study, we have demonstrated that our *emm*-typing tool can confidently call *emm* subtype from WGS data and, in the rare cases were mapping cannot assign *emm* subtype, de novo assembly can be used to determine the *emm* subtype. Although de novo assembly is a useful tool, it does have certain limitations that restrict its usage in a reference laboratory setting. One of the main issues is the lack of local quality metrics that can inform on the area of interest. Even though there are metrics to inform on the quality of the assembly, there is no detailed information for the specific area analysed whereas mapping can offer quality metrics (identity, coverage, depth, mixed bps) for the exact region that is used for each analysis. Furthermore, de novo assembly is time-consuming and considering that in the majority of the cases, the mapping approach was able to successfully assign the correct *emm* subtype it would have been superfluous to implement for all isolates.

In cases where *emm/emm*-like gene pairs were detected, de novo analysis was used to further investigate and confirm their presence. Most *emm*-like genes are localised exclusively before or after the *emm* gene suggesting that based on the *<mrp-emm-enn>* structure they

could be assigned to *mrp* or *enn* respectively. However, *emm* 205.0 and *emm*156.0 seem to alternate positions based on their respective *emm* pair suggesting that the previously suggested structure <*mrp-emm-enn*> is not stable and can vary in some strains.

The automated nature of the *emm* typing tool enables incorporation into routine pipelines and results can be populated into laboratory information systems using custom scripts, thus avoiding potential errors associated with manual result recording and data entry, enhancing reference microbiology.

## ACKNOWLEDGEMENTS

Authors would like to thank Roger Daniel, Chenchal Dhami, Marisa Laranjeira, Timothy Chambers and Sarah Phillips for their technical assistance in performing bacterial purification, DNA extraction, *emm* typing of *Streptococcus pyogenes* strains included in this study. Whole genome sequencing was performed by Catherine Arnold and the team in Genome Services and Development Unit, PHE Colindale. Thanks also go to the CDC streptococcus lab, USA for provision and maintenance the *emm* allele database and all health care scientists and microbiologists for the submission of strains to the reference laboratory.

### Funding

The authors received no funding for this work.

### Competing Interests

The authors declare there are no competing interests.

### Author Contributions

- Georgia Kapatai conceived and designed the experiments, performed the experiments, analyzed the data, contributed reagents/materials/analysis tools, wrote the paper, prepared figures and/or tables.
- Juliana Coelho conceived and designed the experiments, performed the experiments, analyzed the data, contributed reagents/materials/analysis tools, reviewed drafts of the paper.
- Steven Platt conceived and designed the experiments, performed the experiments, contributed reagents/materials/analysis tools, reviewed drafts of the paper.
- Victoria J. Chalker conceived and designed the experiments, contributed reagents/materials/analysis tools, reviewed drafts of the paper.

### Data Availability

Code repository for emm-typing-tool: https://github.com/phe-bioinformatics/emm-typing-tool.

Raw data for the 3047 GAS isolates can be found at the PHE Pathogens BioProject PRJEB17673 at ENA.

## Supplemental Information

Supplemental information for this article can be found online at http://dx.doi.org/10.7717/peerj.3226#supplemental-information.

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
