# Peer review of "Whole genome sequencing of group A Streptococcus: development and evaluation of an automated pipeline for emmgene typing"

_PeerJ, doi:10.7717/peerj.3226_

## Round 0.1 · original submission · Minor Revisions

I have now received two reviews of your paper and both enjoyed the manuscript and thought it a valuable contribution. Reviewer 1 suggests minor revision and reviewer 2 'major revision', but you will see from his review that he actually considers the revision minor (although conceptually a major shift of the emm type definition). I especially like reviewer 1's suggestion to run the data from the Athley et al. paper and then articulate the differences between your approach and this one previously published. Both reviewers make a number of additional comments that I think will help strengthen your paper. Good luck with your revision.

Reviewer 1 ·

Basic reporting

Overall the manuscript is well written and all the data is publicly available and well described.

Experimental design

These kinds of pipelines are useful to those performing large scale whole genome sequencing, such as reference laboratories, and the variation in emm genes do make it difficult to emm type without manual identification, which is not practical on such a scale.

Validity of the findings

As the authors state, Athey et al. have also published a emm-typing tool for WGS. To give readers a better idea of which pipeline to use could the authors be slightly clearer on how their method differs from that already published? Perhaps the authors could test the sequences of Athey et al. in their pipeline to determine if the same or different results are achieved? This would also give a feel for how well the pipeline works on other datasets.

Additional comments

Minor comments to the authors
Methods lines 57-62. Could the authors be slightly clearer on the standard emm-typing here as this is being used as a comparison and it is not 100% clear how this is performed
Line 91. What do you mean here by ‘validated’?
Lines 132-143. Sounds a bit like WGS is able to pick up on multiple subtype variants within patients? This is an interesting observation that could be followed up on as it is not clear if there are multiple sub-types within patients.
Lines 153-158. Did this happen with certain emm-types or was it a random observation?
Lines 217-219. Can the authors clarify this sentence slightly as I do not understand it.
Line 222. The use of the word transcription is confusing here as it is common to associated it with transcription of genes rather than mis-labelling of samples!

·

Basic reporting

Don't get me wrong. I like the work. That said, I have recently come to the conclusion that the field is making the emm type definition more complicated than it needs to be. Please consider what I have to say and simplify your paper a bit (and please don't consider me a hypocrite due to my previous coauthorship (Athey et al). Since this paper my lab has emm typed many thousands of GAS ourselves using Illumina sequencing (with de novo assembly approach!), and I believe that I see the light!

1. Nice work, but I will cut to the chase. When using a short read sequencing approach one should use solely denovo assembly and NOT read mapping- otherwise the data will be occasionally confounded by emm-like gene sequence. This is the crux of my review. Feel free to contact me directly.
2. I realize that I currently have contradictory information regarding the emm type definition at the CDC web site (and now you know who I am- tis ok!). I am going to change it back to this: The “emm type” definition solely should be simply the sequence that is physically linked to the specific primer 1 PCR/sequencing primer (as defined at the CDC web site). This primer practically always is linked to the actual M protein gene that confers the M serotype, but this is not the entire point (as an aside, one must be very careful if thinking this primer can be improved due to sequence similarities at neighboring mrp and enn genes). I also realize that I am contradicting a paper (Athey et al.) on which I coauthored. Since this time, we have implemented WGS as our sole means of typing GAS. I now see the error of using read mapping for determining emm type!

3. line 45. I don’t agree with extending the simple primer 1 based emm type to “validation of the M-antigen”. The emm type is a simple typing tool and should not be made more complex. WHile the primer 1 defined emm type almost always does correlate to an actual M serotype, one cannot assume this. The only validation that should be required is that the specific primer 1 sequence maps closely upstream (within the signal peptide-encoding sequence actually) of the emm type-specific sequence. If this is not a requirement, then the CDC emm type definition is not being followed. There is good reason that this simple approach should be followed in that the emm-like genes that flank the so-called emm locus (the one preceded by the primer 1 anealing sequence) have closely similar regions in their respective signal sequence encoding regions.

4. line 178. Don't worry about emm-like genes in the database. They are rarely encountered and I admit that in many instances they are probably due to people trying to hard to get amplicons using the emm typing primers. Other times, they are legitimate emm types due to intra-strain recombination events. At any rate, such instances are rare and if emm typing is done correctly (either classical method or WGS with de novo assembly) these database entries will not contaminate your surveillance data! Bottom line is that the rare oddities reflect a global sequence-based database. Those emm types that are not relevant will just sit in the database and will not adversely affect surveillance data.

5. Please acknowledge our CDC emm database—and thanks!

Experimental design

Nice work, but I will cut to the chase. When using a short read sequencing approach one should use solely denovo assembly and NOT read mapping- otherwise the data will be occasionally confounded by emm-like gene sequence. This is the crux of my review. Feel free to contact me directly.

Validity of the findings

All the findings are valid, but some should be re-interpreted.

Additional comments

Say, I did say "Major Revisions" but actually it is a major concept that I believe that should be brought out that at the same time would be pretty simple for you to make clear. My suggestion to rely solely upon de novo assembly (in such a way to link emm subtype with primer 1) would simplify the paper and greatly improve its logic to this old strep typer.

Thanks for using the database, but take the "emm-like" genes with a grain of salt. THey are only legit if in fact linked to the primer 1 sequence.

I need to change some of the dialogue at https://www.cdc.gov/streplab/m-proteingene-typing.html where I discuss the paper by Athey et al. I do not quite buy-in to it now.

Best Regards,
Bernie

---

## Round 0.2 · Minor Revisions

Just a few more minor edits and I think we'll be there. Please see these last issues from Reviewer 1. Thanks.

Reviewer 1 ·

Basic reporting

There are some mistakes in the abstract - for example I think there is a word missing in the second line. There are also several instances throughout the manuscript where emm needs to be italicised. Can the authors please also change the sentence in line 380 to remove the use of the word 'you'.

Experimental design

No comment

Validity of the findings

No comment

Additional comments

I feel that the authors have addressed all my comments appropriately.

---

## Round 0.3 · accepted · Accept

Thanks for your careful revision. I think we are good to go now!